# Urinary Zinc Loss Identifies Prostate Cancer Patients

**DOI:** 10.3390/cancers14215316

**Published:** 2022-10-28

**Authors:** Maria Grazia Maddalone, Marco Oderda, Giulio Mengozzi, Iacopo Gesmundo, Francesco Novelli, Mirella Giovarelli, Paolo Gontero, Sergio Occhipinti

**Affiliations:** 1Clinical Biochemistry Laboratory, Department of Laboratory Medicine, AOU Città della Salute e della Scienza di Torino, 10126 Turin, Italy; 2Division of Urology, Department of Surgical Science, AOU Città della Salute e della Scienza di Torino, University of Turin, 10124 Turin, Italy; 3Division of Endocrinology, Diabetes and Metabolism, Department of Medical Sciences, University of Turin, 10124 Turin, Italy; 4Department of Molecular Biotechnologies and Health Sciences, University of Turin, 10126 Turin, Italy; 5NIB biotec srl, 10135 Torino, Italy

**Keywords:** prostate cancer prevention, prostate cancer detection, screening, biomarkers, diagnosis, early detection

## Abstract

**Simple Summary:**

Prostate cancer is known to lose the capability to absorb and secrete zinc compared to normal prostate tissue, suggesting that the evaluation of zinc in prostate secretion can be a tool to identify the risk of developing cancer. In our study, we observed that the average amount of zinc detectable in urine after a prostatic massage is lower in patients with prostate cancer than in healthy subjects. Moreover, there is an inverse correlation between the concentration of urinary zinc and the tumor stage. This evidence suggests that the evaluation of urinary zinc may be a parameter for better diagnosis and prognosis of prostate cancer.

**Abstract:**

Prostate Cancer (PCa) is one of the most common malignancies in men worldwide, with 1.4 million diagnoses and 310,000 deaths in 2020. Currently, there is an intense debate regarding the serum prostatic specific antigen (PSA) test as a diagnostic tool in PCa due to the lack of specificity and high prevalence of over-diagnosis and over-treatments. One of the most consistent characteristics of PCa is the marked decrease in zinc; hence the lost ability to accumulate and secrete zinc represents a potential parameter for early detection of the disease. We quantified zinc levels in urine samples collected after a standardized prostatic massage from 633 male subjects that received an indication for prostate biopsy from 2015 and 2019 at AOU Città della Salute e della Scienza di Torino Hospital. We observed that the mean zinc levels were lower in the urine of cancer patients than in healthy subjects, with a decreasing trend in correlation with the progression of the disease. The combination of zinc with standard parameters, such as PSA, age, digital rectal exploration results, and magnetic resonance findings, displayed high diagnostic performance. These results suggest that urinary zinc may represent an early and non-invasive diagnostic biomarker for prostate cancer.

## 1. Introduction

The diagnostic process to detect prostate cancer (PCa) includes measuring serum prostatic specific antigen (PSA) levels and a digital rectal examination (DRE). When PCa is diagnosed early, the treatment can be effective and with minimal morbidity [1], underlying the key role of urological consultations.

Currently, there is an intense debate regarding the serum PSA test as a screening tool for PCa due to the lack of specificity and high prevalence of over-diagnosis and over-treatments [2]. It is, therefore, important to focus on other molecular markers that can be of aid in diagnostic and therapeutic management [3]. 

In particular, the controversial PSA screening sustains the need for alternative biomarkers with better diagnostic and predictive potential, capable of distinguishing between aggressive and indolent cancers. In the last years, several biomarkers have been discovered, suggesting better performances compared to PSA. Among these, we cite the Prostate Health Index (PHI) [4], 4K score [5], SelectMDx [6], and PCA3 [7].

However, all these new approaches led only to limited benefits in the diagnostic-therapeutic pathway of PCa, mainly because a unique gene signature cannot be universally applicable to all patients with an oncologic disease. 

A shared aspect that characterizes neoplastic transformation is a profound change at both phenotypic and functional levels. One of the physiological functions of the prostate is to produce part of the fluid that forms semen. The majority of the prostate is composed of glandular cells; hence the most common type of cancer that affects this organ (i.e., adenocarcinoma) can impair the composition of the prostate fluid. The amount of several products of the prostate, such as PSA is decreased in advanced cancer [8,9], and their evaluation in prostatic secretion or urine can be used to identify the presence and the stage of the disease [10].

Together with the mammary and pancreatic tissues, the prostate is one of the organs that physiologically accumulate most of the zinc circulating in our body [11]. Zinc inside the prostate acts in the mitochondria by inhibiting the oxidation of citrate to isocitrate [12], and the accumulated citrate has the function of promoting the release of sperm [13].

Prostate cells actively accumulate zinc. Several studies have shown that damage to the functionality of the cells, for example, due to a neoplastic transformation, leads to a loss of the ability to absorb zinc and therefore, to a lower expression in the tissue [14,15,16].

In this study, the level of zinc present in the urine of men who received an indication to undergo a prostate biopsy for a suspected tumor was quantified. The amount of zinc was found to be lower in the urine of patients with cancer than those with negative biopsy. Also, the average urinary zinc level gradually decreases as the disease progresses. These results are in line with what has already been observed in the tissue. However, here we reported for the first time that urinary zinc can be used for the diagnosis and prognosis of PCa, alone and in combination with standard parameters such as PSA or multiparametric Magnetic Resonance (mpMRI).

## 2. Materials and Methods

### 2.1. Study Population and Study Design

Men scheduled for prostate biopsy at the urology unit in the AOU Città della Salute e della Scienza di Torino (Turin, Italy) from June 2015 to May 2019 (four years) were invited to participate in the study. The indication for prostate biopsy was defined according to routine clinical parameters, such as PSA levels, digital rectal exploration (DRE), and MRI imaging. Before the biopsy, we collected a urine sample from each subject after a standardized prostatic massage. Participants (n = 633) were divided into Training Cohort 1 (n = 411) and Validation Cohort 2 (n = 212). Subjects with PSA levels above 25 ng/mL (n = 38) were excluded from the analysis. 

Subjects that received a diagnosis of prostate cancer (PCa patients) were stratified according to PSA, Gleason Score (GS), and tumor staging [17] into low risk (PSA < 10 ng/mL and GS < 7 (ISUP 1) and cT1-2a), favorable-intermediate risk (PSA 10–20 ng/mL or GS 7 (ISUP 2) or cT2b), unfavorable-intermediate (ISUP 3, or > 50% positive biopsy cores or at least two favorable-intermediate-risk factors), and high risk PCa (PSA > 20 ng/mL or GS > 7 (ISUP 4/5) or cT2c) [17]. A PCa with ISUP >1 was considered “clinically significant” (csPCa).

A urine sample was collected from 78 men with an indication of prostate biopsy without prostate massage prior to collection.

### 2.2. Sample Collection and Processing

Urine samples were collected and processed as previously described [10]. 

A standardized prostate massage was performed to extract prostatic secretions through three digital compressions in each lobe, starting at the base, moving down to the center and apex in a 30-s time lapse.

Approximately 30 mL of first voided urine were collected after gentle agitation, a 15 mL aliquot was stored in plastic tubes at −80 °C within 5 min of collection.

### 2.3. Urine Analysis

Urinary zinc measurement was performed in the Atomic Absorption Laboratory (Baldi e Riberi), AOU Città della salute e della Scienza di Torino, Turin, Italy.

Samples were first acidified with 100 μL 18% HCl and then diluted 1: 5 with CsLaCl 1 g/L (1 mL sample + 4 mL CsLaCl 1 g/L). They were then analyzed by the flame atomic absorption technique. In addition, four calibration curve standards were used for the analysis (S1: 4.5 mL CsLaCl 1 g/L + 400 μl H2O + 100 μL Std Cu/Zn 1 mg/L; S2: 4.5 mL CsLaCl 1 g/L + 500 μL Std Cu/Zn 0.5 mg/L; S3: 4.5 mL CsLaCl 1 g/L + 500 μL Std Cu/Zn 1 mg/L; S4: 4.5 mL CsLaCl 1 g/L + 500 μL Std Cu/Zn 2 mg/L), and two controls (Seronorm Urine L-1 and Seronorm Urine L-2) diluted 1: 4 with CsLaCl 1 g/L (1 ml control + 3 ml CsLaCl 1 g/L).

Once prepared, the samples were analyzed on the Zeenit 700p instrument (flame atomic absorption spectrometer, Analytik Jena).

### 2.4. Ethics Statement

A human investigation was conducted according to the Declaration of Helsinki principles after the approval of the study by the Scientific Ethics Committee of AOU Città della Salute e della Scienza di Torino, A.O. Mauriziano, A.S.L. TO1 (Prot. No. 0110644). Each participant released written informed consent before inclusion in the study, and specimens were anonymized after collection.

### 2.5. Study Endpoints and Statistical Analyses

We categorized subjects by risk class based on PSA value, GS, the number of positive biopsies, and TNM staging [18]. 

We performed univariate and multivariate logistic regression analysis to evaluate the diagnostic performance of urinary zinc in detecting csPCa, alone or in combination with other parameters, such as PSA level, DRE finding, age, and mpMRI. We designed different multivariate logistic models, and the discriminative power was assessed by calculating the area under the receiver operating characteristic (ROC) curves (AUC). We compared the diagnostic performance of four different multivariate logistic regression models. The first model (PSA model) included serum PSA level only. The second model (SOC model) included PSA level, age at diagnosis, and abnormal DRE. The third model (Zinc model) included urinary zinc level. The fourth model (Zinc + SOC model) included urinary zinc level, serum PSA level, age, and abnormal DRE. The fifth model (MRI model) considered PiRADS value. The sixth (SOC + MRI model) included serum PSA level, age, abnormal DRE, and PiRADS. The seventh (Zinc + SOC + MRI model) included urinary zinc, PSA level, age, abnormal DRE, and PiRADS. Comparisons of AUCs of different models were determined using DeLong’s method. Logistic regression coefficients were estimated in Cohort 1 and externally validated in Cohort 2.

The Number Needed to Predict (NNP), representing the number of patients who need to be examined in a population to correctly predict the diagnosis of one person, was also calculated [19].

Statistical analyses were performed as previously published [9] with MedCalc^®^ Statistical Software version 19.8 (MedCalc Software Ltd., Ostend, Belgium).; https://www.medcalc.org; 2021

## 3. Results

### 3.1. Patient Characteristics

Recruited men were divided into training Cohort 1 and validation Cohort 2. In Cohort 1 (n = 394), 187 men had a negative biopsy (100 with no evidence of a tumor or with benign prostatic hyperplasia (BPH), 51 inflammation/prostatitis, 21 high-grade prostatic intraepithelial neoplasia, and 15 atypical small acinar proliferation), while 207 (49.7%) had a positive biopsy outcome.

Based on D’Amico’s stratification, 17 patients had low risk, 68 favorable-intermediate (int fav) risk, 67 unfavorable-intermediate (int unfav) risk, and 55 high risk PCa. The percentage of clinically significant (cs) PCa was 48.2% (Table 1).

For Cohort 2, of 201 patients, 74 had a negative biopsy and 127 (63.2%) a positive biopsy outcome, of which 12 were low risk, 38 int fav risk, 38 int unfav risk, and 39 high risk PCa. The prevalence of csPCa was 57.2% (Table 1).

The average age of subjects was 68 years in both Cohorts. In Cohort 1 and Cohort 2, there was no difference in terms of mean PSA value (7.2 vs. 7.8 ng/)mL. Subjects in Cohort 1 displayed higher DRE abnormality compared to Cohort 2 (*p* = 0.02) but a lower prevalence of csPCa (*p* = 0.01).

### 3.2. Quantification of Urinary Zinc in Subjects Candidate for Prostate Biopsy

Urine samples collected before prostate biopsy were tested for the presence of zinc. We observed a gradual decrease in zinc levels at increasing PCa class risk (*p* for trend 0.0001, Figure 1, Table 2).

The mean levels of zinc were significantly lower in patients with Int-fav, Int-unfav, and high risk PCa (0.66, 0.65, 0.60 µg/mL, respectively) compared to healthy subjects (1.02 µg/mL) and low-risk PCa (1.93 µg/mL). No differences were observed between low-risk PCa and healthy subjects (Figure 1, Table 2). 

To evaluate the role of prostate massage in extracting prostate contents, zinc was evaluated in the urine collected from men with an indication of biopsy without performing this procedure. In the absence of prostate massage, the average zinc level is lower than that observed in the presence of massage, with no significant differences between healthy individuals and prostate cancer patients (Appendix A).

### 3.3. Evaluation of Urinary ZINC and Routine Parameters 

We compared the levels of urinary zinc in subjects without evidence of PCa or low-risk PCa (non-cancer) to that of patients with csPCa. 

Urinary zinc levels were significantly lower in csPCa patients than in non-cancer individuals (Figure 2A; *p* = 0,0001). The average age (Figure 2C) and percentage of suspect DRE (Figure 2D) were significantly higher in csPCa patients compared to non-cancer individuals, with a *p*-value of 0.0087 and 0.045, respectively. No differences were observed in the average PSA value (Figure 2B).

No significant correlation was observed between urinary zinc, PSA, age, or DRE (Figure 2E–G).

This evidence suggests that lower zinc values are detectable in the urine of patients with prostate cancer than in healthy subjects and that the decrease in urinary zinc is an independent indicator of standard parameters.

### 3.4. Diagnostic Accuracy of Urinary Zinc Levels to Identify Clinically Significant Prostate Cancer

We evaluated the capability of urinary zinc analysis in discriminating non-cancer individuals from csPCa patients. 

Table 3 shows the results of the four logistic regression models in detecting PCa. Increased probability of zinc and zinc + SOC associated with prostate cancer (OR for unit increase 2.20 and 3.21, respectively).

The AUC for blood PSA, the standard of care parameters (PSA, age, and DRE; SOC), urinary zinc alone, or zinc + SOC was 0.551, 0.607, 0.652, and 0.687, respectively. The AUC of both zinc alone and zinc + SOC was significantly higher than the AUC of PSA (Table 3 and Figure 3: *p* = 0.0143 and *p* = 0.0002, respectively). Furthermore, the AUC of zinc + SOC was also higher than the AUC of SOC alone (*p* = 0.0011).

At a sensitivity of 95% or 90%, the specificity for zinc and zinc + SOC were greater than PSA alone or SOC.

At 95% of sensitivity, zinc alone or the combination of zinc + SOC showed higher Positive predictive value (PPV) and negative predictive value (NPV), hence greater capacity in accurately classifying biopsies results compared to PSA alone and SOC (Table 4).

At 95% sensitivity, the Number Needed to Predict (NNP) was lower for zinc and zinc + SOC than for PSA and SOC (Table 4).

We evaluated if zinc had different diagnostic capabilities in different subgroups of subjects based on PSA value or age. In subjects with a PSA range between 0 and 4 ng/mL, 4.1–10 ng/mL, and 10.1–25 ng/mL, analysis for Zinc levels showed AUCs of 0.589, 0.642, and 0.753, respectively (Table 5).

The AUC of zinc for patients aged below 60, between 60 and 75, and above 75 were 0.629, 0.637, and 0.697, respectively (Table 5).

### 3.5. External Validation of the Diagnostic Models

Four diagnostic models based on PSA, age, DRE, and urinary zinc were developed in Cohort 1 for the capability to assess the probability of csPCa. 

The diagnostic performance of these models was validated in an independent group of men candidates for prostate biopsy (Cohort 2). The AUC for PSA alone, SOC, zinc, and zinc + SOC in predicting the chance of csPCa was 0.558, 0.669, 0.683, and 0.735, respectively (Table 6). The AUC for SOC, zinc, and zinc + SOC were significantly higher than that of PSA alone with *p* = 0.0085, *p* = 0.0195, and *p* = 0.0001, respectively. The combination of zinc + SOC showed higher AUC (*p* = 0.0177) compared to SOC (Table 6 and Figure 4).

### 3.6. Urinary zinc and Multiparametric Magnetic Resonance Imaging

As mpMRI is becoming the gold standard for prostate biopsy candidate selection, we investigated whether urinary zinc assessment could increase the diagnostic capacity of mpMRI and SOC. In individuals undergoing prostate biopsy for suspected mpMRI (*n* = 226), 82 were disease-free, 11 had low-risk PCa, 49 Int-fav PCa, 49 Int-unfav PCa, and 35 high-risk PCa.

The combination with SOC does not increase the diagnostic performance of mpMRI in detecting csPCa (Figure 5 and Table 7). The combination of mpMRI with SOC and zinc demonstrated significantly superior diagnostic performance than SOC and mpMRI alone or in combination (Table 7).

The mpMRI results in combination with SOC showed similar diagnostic performance in detecting csPCa compared to mpMRI alone (Figure 5 and Table 7). The combination of mpMRI with SOC and zinc demonstrated a diagnostic performance significantly superior to SOC and mpMRI alone or in combination (Table 7).

Furthermore, we separately evaluated the diagnostic performance of zinc and SOC in subjects with Prostate Imaging Reporting and Data System (PiRADS) values of 3, 4, and 5. In PiRADS groups 3 and 4, the combination of zinc + SOC displayed an AUC of 0.827 and 0.730, which are statistically different compared to PSA alone (Table 8, *p* = 0.009 and *p* < 0.0001, respectively).

In PiRADS group 5, the combination of zinc + SOC displayed an AUC of 0.835; that, however, is not statistically different from PSA alone (*p* = 0.1).

### 3.7. Prostate Cancer Risk Probability Combining Zinc with MRI and Standard Parameters

The clinical utility of a diagnostic biomarker for prostate cancer should be defined by calculating the potential capacity to reduce the use of biopsy in individuals without prostate cancer. The probability that each subject has of having prostate cancer was calculated based on the zinc + SOC + MRI diagnostic model. We observed that considering subjects with a probability of having a csPCa greater than 30% as suspect; 28% of unnecessary biopsies could have been avoided without missing any high-grade cancers.

With a probability of more than 40%, a potential 40% reduction in unnecessary biopsies is achieved, losing only 6% of high-grade cancers (Table 9).

### 3.8. Diagnostic Accuracy of Urinary Zinc Levels Evaluation in Patients Undergoing Repeated Prostate Biopsy

To assess the impact of the use of urinary zinc analysis as a tool on the repeated biopsy decision-making process, the SOC and zinc models were applied to subjects who had already undergone a previous biopsy with a negative outcome that received an indication to perform a further biopsy for a permanent suspicion. Of 82 patients, 40 had no evidence of PCa, and 42 were diagnosed with csPCa.

As shown in Table 10, the AUC of zinc + SOC (0.764) was significantly higher than the AUC of PSA and SOC alone (*p* = 0.002 and *p* = 0.009, respectively).

## 4. Discussion

The discovery and validation of new biomarkers to avoid unnecessary biopsies and reduce the risk of overdiagnosis and overtreatment for PCa is urgently needed in this field.

A multitude of new diagnostic tools have emerged in recent years and have become available on the PCa market, with the intention of providing more accurate information than the standard PSA and DRE test on the actual risk of cancer [20].

Although several tests have shown a potential ability to improve the diagnostic and therapeutic pathway, there is a lack of prospective studies to support their impact on disease outcomes.

As biomarkers become available, it is increasingly important to understand how, when, and on which patients they should be used. Integration into routine clinical practice and insurance coverage are still areas where more work needs to be done for the benefit of both doctors and patients.

However, the latest update of the European guidelines of the European Association of Urology (EAU-ESTRO-SIOG) recommended none of these tests or tools in support of the standard of care [21]. 

To date, with the technological advances in MRI, with improvements in the quality and standardization of interpretations with PIRADS-v2 [22], mpMRI has taken its place at the forefront of PCa detection, being recommended by the AUA and UAE guidelines in suspected patients to have PCa [22,23]. Despite its valuable role in PCa diagnosis, however, mpMRI is far from flawless and a clinically significant portion of PCa is still lacking. Therefore, the addition of a diagnostic biomarker other than PSA to the mpMRI could represent a further improvement in the diagnostic setting of PCa.

The results of this study highlighted that a significant decrease in mean zinc levels is observable in the urine of PCa patients compared to healthy subjects. This evidence is in line with previous studies showing that the level of zinc in prostate tissue and prostatic secretion is substantially lowered in the presence of neoplastic transformation compared to non-malignant pathologies and normal glands [24,25,26].

Urinary zinc showed an additive value in combination with standard of care and mpMRI. Available evidence suggests that the incorporation of mpMRI in the diagnostic pathways for males should be recommended. 

However, despite a bought advantage in the use of this approach in the diagnostic path of prostate cancer, the need for instrumentation, specially trained healthcare personnel, and the high costs greatly limit its use as a mass screening method [27].

Furthermore, the high probability that clinically significant prostate cancer will not be detected on MRI leaves many concerns about its use as a first-line test [28]. The latest meta-analysis showed that the NPP of mpMRI ranges from 80% to 95% [29], indicating a considerable risk of losing csPCa, with a probability of false positive results varying from 10% to 50% even for high PiRADS values [30].

In this study, we observed that urinary zinc analysis, mpMRI imaging results, and evaluation of standard parameters are complementary. Their simultaneous evaluation could potentially improve accuracy in identifying the correct candidates for prostate biopsy, thus reducing the risk of false positive and false negative results.

Several findings provide evidence that strategies combining blood or urine biomarker testing and MRI can improve clinical routine by better detecting prostate cancer early and providing insight for biopsy decision-making [31].

Interestingly, in vivo studies in mice showed that zinc detection by imaging technologies can provide a specific diagnostic method to differentiate non-cancerous prostate tissue from prostate cancer in situations where it may be difficult to detect using current multiparameter MRI protocols [32].

The main limitation of our study is that not all the subjects underwent mpMRI for biopsy decision-making. In further studies, it would be interesting to assess the diagnostic performance of urinary zinc, alone or in combination with other parameters or biomarkers, in a wide range of subjects in different settings, such as symptomatic or general populations undergoing the current diagnostic path, including state-of-the-art magnetic resonance imaging.

Future studies will also be performed on men who have not received a prostate biopsy indication, comparing the values of urinary zinc with the results of MRI.

The prostatic gland is one of the tissues that particularly accumulate zinc for biological processes. The alteration of zinc absorption and accumulation is one of the key characteristics of PCa. During the progression of the disease, lower zinc levels can be found in comparison to normal tissue. 

Moreover, PCa appears to be the only prostatic disease in which zinc accumulation is compromised. Other studies highlighted that the decrease in zinc levels is not a characteristic of BPH or prostatitis [33], suggesting that urinary zinc analysis can efficiently discriminate patients with PCa from other conditions, representing an excellent candidate biomarker.

Urinary zinc measurement is already in use in some clinical biochemistry laboratories to detect industrial and accidental exposure to zinc or malabsorption. This indicates that the evaluation of urinary zinc is already in use and therefore, there is optimized and dedicated instrumentation and sample preparation methods to carry out this evaluation.

PCa is highly heterogeneous, mainly multifocal, and difficult to be fully characterized with a unique signature. It may therefore be necessary to identify and clinically validate a panel of biomarkers that evaluate different aspects of tumor progression to better select suitable patients for prostate biopsy.

## 5. Conclusions

In conclusion, the loss of zinc accumulation and secretion by the prostate during neoplastic transformation could potentially represent a hallmark of PCa, and its combination with standard diagnostic parameters and mpMRI could represent an interesting approach in the diagnosis of PCa.

Although the present study showed a better diagnostic capacity of PCa with the implementation of urinary zinc, results suggest space for improvement, potentially in combination with other urinary biomarkers with similar physiology.

## Figures and Tables

**Figure 1 cancers-14-05316-f001:**
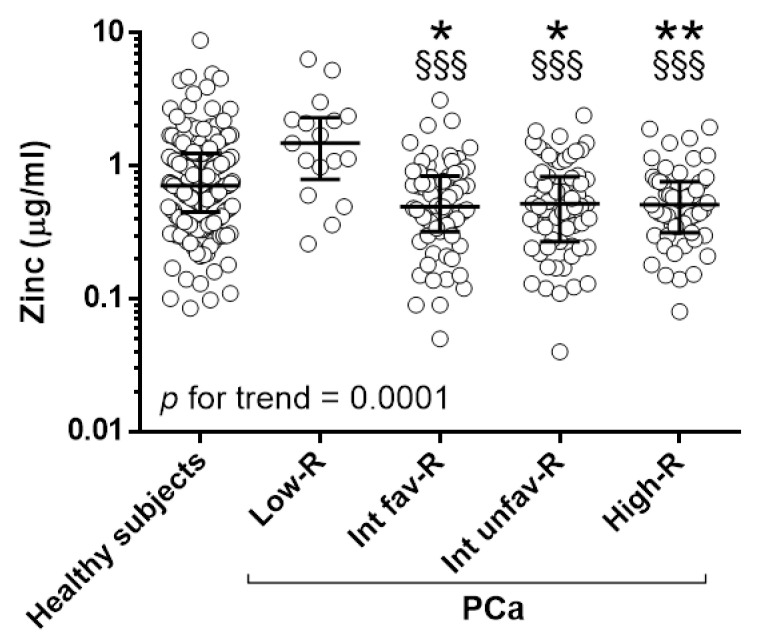
Urinary zinc in healthy individuals and patients with PCa: Zinc in men with no evidence of PCa (Healthy subjects, *n* = 187), low- (Low-R, *n* = 17), intermediate-favorable- (Int-fav-R, *n* = 68), intermediate-unfavorable (Int-unf-R, *n* = 67), high- (High-R, *n* = 55) risk patients. * *p* < 0.05, ** *p* < 0.01, to healthy subjects. §§§ *p* < 0.001 to Low-R. *p* for the trend = 0.0001.

**Figure 2 cancers-14-05316-f002:**
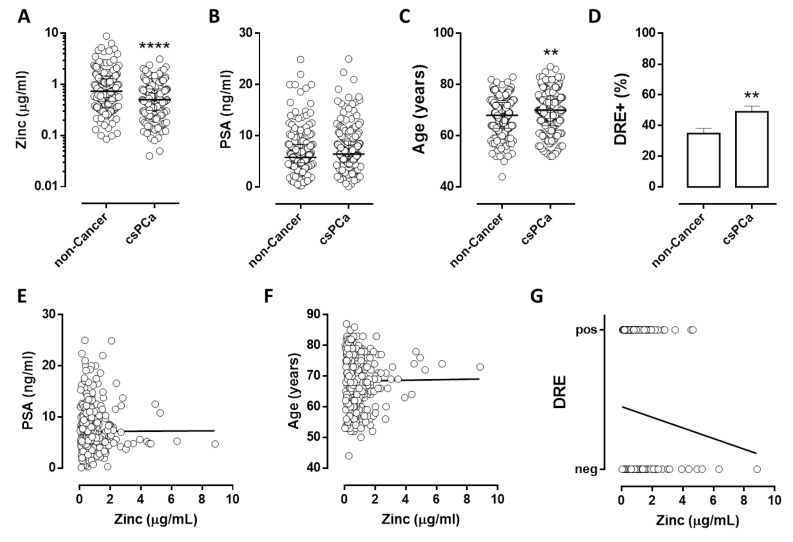
Urinary zinc and standard parameters in patients with clinically significant PCa: amount of zinc (**A**), serum PSA (**B**), Age (**C**), percentage of suspect DRE (**D**) in men without prostate cancer or Low-risk PCa (non-cancer, *n* = 204) and patients with clinically significant PCa (csPCa, *n* = 190). Correlation between zinc and standard parameters: PSA (**E**), Age (**F**), and DRE (**G**) in subjects candidate for prostate biopsy. ** *p* < 0.01, **** *p* < 0.0001, to non-cancer.

**Figure 3 cancers-14-05316-f003:**
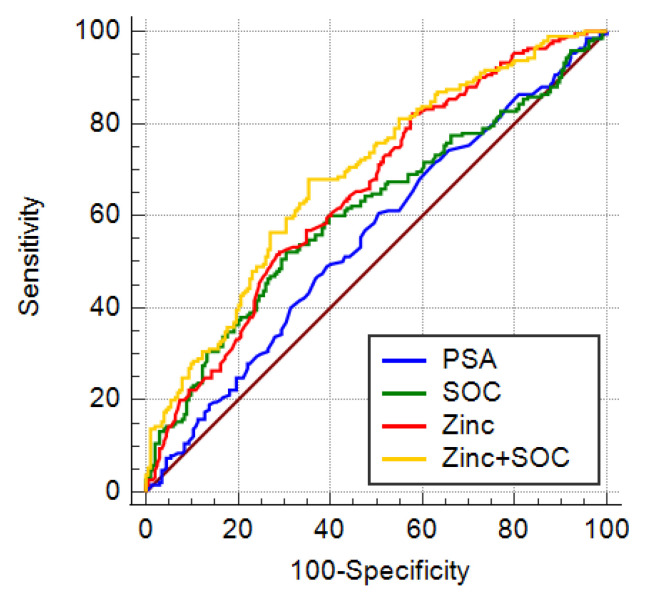
Diagnostic performance in detecting csPCa: AUC ROC for standard parameters PSA (blue line), SOC (PSA, age, DRE, green line), Zinc (red line), combination of zinc and SOC (Zinc + SOC, yellow line).

**Figure 4 cancers-14-05316-f004:**
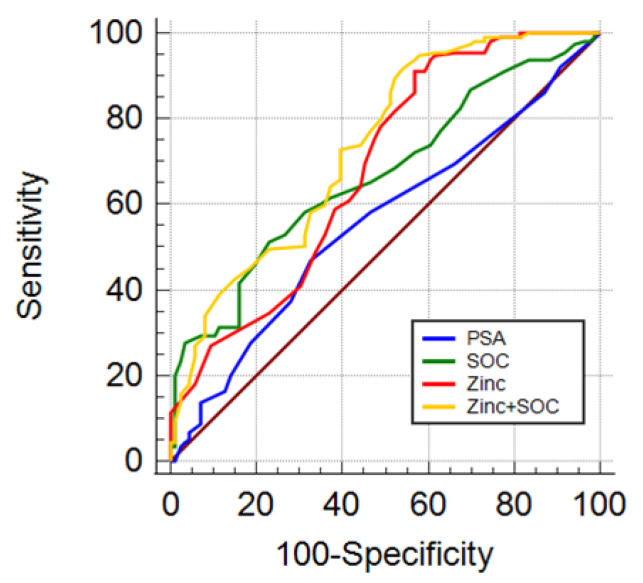
Diagnostic performance in detecting csPCa: AUC ROC for standard parameters PSA (blue line), SOC (PSA, age, DRE, green line), Zinc (red line), combination of zinc and SOC (zinc + SOC, yellow line).

**Figure 5 cancers-14-05316-f005:**
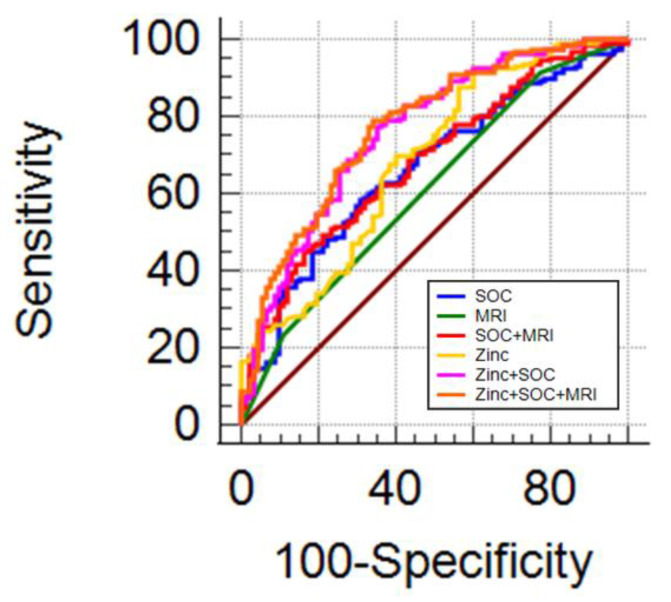
Diagnostic performance of zinc and MRI: AUC ROC for SOC (blue line), MRI (green line), SOC + MRI (red line), zinc (yellow line), zinc + SOC (pink line), zinc + SOC + MRI (orange line).

**Table 1 cancers-14-05316-t001:** Patient characteristics.

Characteristics	Cohort 1	Cohort 2	*p*
Patients, *n*	411	212	-
Evaluable samples, *n* (%) ^1^	394 (96)	201 (95)	-
Age, yr, mean (median; IQR)	68 (69; 63–74)	68 (69; 63–73)	ns ^2^
PSA, ng/mL mean (median; IQR)	7.2 (6.1; 4.8–8.6)	7.8 (6.6; 4.7–9.6)	ns
DRE abnormal, *n* (%)	164 (41.6)	64 (31.8)	0.02
PCa diagnosis, *n* (%)	207 (52.5)	127 (63.2)	0.01
Low risk, *n* (%)	17 (8.2)	12 (9.4)	-
Intermediate favorable risk, *n* (%)	68 (32.9)	38 (29.9)	-
Intermediate unfavorable risk, *n* (%)	67 (32.4)	38 (29.9)	-
High risk, *n* (%)	55 (26.6)	39 (30.7)	-
Clinically significant PCa, *n* (%)	190 (48.2)	115 (57.2)	0.04

^1^ Number of evaluable samples based on a serum PSA < 25 ng/mL ^2^ Not significant.

**Table 2 cancers-14-05316-t002:** Urinary zinc expression among healthy subjects and patients.

Diagnosis	Mean (µg/)mL	Median (p25-p75)	*p*	
Healthy subjects	1.02	0.71 (0.45–1.24)	ref ^1^	-
Low Risk	1.93	1.48 (0.79–2.31)	ns ^2^	ref
Int-fav ^3^ Risk	0.66	0.50 (0.32–0.84)	0.0163	0.0004
Int-unfav ^4^ Risk	0.65	0.52 (0.27–0.83)	0.0139	0.0004
High Risk	0.60	0.51 (0.32–0.76)	0.0080	0.0002
*p* for trend	<0.0001			

^1^ reference; ^2^ not significant; ^3^ Intermediate favorable; ^4^ Intermediate unfavorable.

**Table 3 cancers-14-05316-t003:** Odds ratios, 95% confidence intervals for PCa.

Model	OR ^1^	95% CI ^2^
PSA	1.35	0.836–2.172
SOC	2.39	1.582–3.608
Zinc	2.20	1.465–3.300
Zinc + SOC	3.21	2.125–4.845

^1^ Odds Ratio, ^2^ Confidence interval.

**Table 4 cancers-14-05316-t004:** Diagnostic models and accuracy.

Model	AUC ^1^	SE ^2^	95% CI ^3^	*p*		Spec ^4^	Spec ^5^	PPV ^6^	NPV ^7^	NNP ^8^
PSA	0.551	0.0290	0.501–0.601	ref	-	7.5	11.3	49.1	64	7.6
SOC	0.607	0.0287	0.557–0.656	ns	ref	7.5	11.3	49.2	66.7	6.3
Zinc	0.652	0.0274	0.602–0.699	0.0143	ns	19	32.8	52.6	82	2.9
Zinc + SOC	0.687	0.0265	0.639–0.733	0.0002	0.0011	23.4	31.8	51.3	78	3.4

^1^ Area under the curve, ^2^ Standard error, ^3^ Confidence interval, ^4^ Specificity at 95% sensitivity, ^5^ Specificity at 90% sensitivity, ^6^ Positive predictive value, ^7^ Negative predictive value, ^8^ Number needed to predict.

**Table 5 cancers-14-05316-t005:** Urinary zinc performance in patients’ subgroups.

Group	AUC ^1^	SE ^2^	Spec ^3^
PSA ≤ ≤4	0.589	0.0778	21.21
4 < PSA ≤ 10	0.642	0.034	16.79
PSA > 10	0.753	0.0569	26.32
Age ≤ 60	0.629	0.0705	11.21
60 < Age ≤ 75	0.637	0.0346	21.58
PSA > 75	0.697	0.0615	26.32

^1^ Area Under the Curve; ^2^ Standard Error; ^3^ Specificity at 95% of Sensitivity.

**Table 6 cancers-14-05316-t006:** Validation of diagnostic models.

Model	AUC	SE	95% CI	*p*		Spec
PSA	0.558	0.0406	0.487–0.628	ref	-	6.1
SOC	0.669	0.0377	0.599–0.734	0.0085	ref	9.5
Zinc	0.683	0.0387	0.614–0.747	0.0195	ns	37.2
Zinc + SOC	0.735	0.0357	0.668–0.795	0.0001	0.0177	41

**Table 7 cancers-14-05316-t007:** Diagnostic performance of zinc and MRI.

Model	AUC	SE	95% CI	*p*		
SOC	0.655	0.0366	0.599–0.727	ref	-	-
MRI	0.609	0.0314	0.542–0.674	ns	ref	-
SOC + MRI	0.684	0.0357	0.618–0.744	ns	0.0135	ref
Zinc	0.685	0.0366	0.620–0.746	ns	ns	ns
Zinc + SOC	0.761	0.0330	0.699–0.815	0.0014	0.0004	0.0197
Zinc + SOC + MRI	0.773	0.0320	0.712–0.826	0.0007	0.0001	0.0017

**Table 8 cancers-14-05316-t008:** Diagnostic performance in MRI subgroups.

PiRADS	PSA	SOC	Zinc	Zinc + SOC
3	0.455	0.779 *	0.680	0.827 **
4	0.456	0.568	0.723 ****	0.730 ****
5	0.652	0.798	0.563	0.835

* *p* < 0.05, ** *p* < 0.01, **** *p* < 0.0001, compared to PSA.

**Table 9 cancers-14-05316-t009:** Clinical performance of urinary zinc.

Cut-Off (Probability)	All N (%)	Non-Cancer N (%)	csPCa N (%)	Missed High Risk N (%)	Saved Unnecessary Biopsies N (%)
0	226 (100)	93 (100)	133 (100)	0 (0)	0 (0)
25	201 (89)	71 (76)	130 (98)	0 (0)	22 (24)
30	196 (87)	67 (72)	129 (97)	0 (0)	26 (28)
40	178 (79)	56 (60)	122 (92)	2 (6)	37 (40)
45	167 (74)	51 (55)	116 (87)	3 (9)	42 (45)
50	152 (67)	42 (45)	110 (83)	5 (14)	51 (55)

**Table 10 cancers-14-05316-t010:** Diagnostic performance in repeated biopsy.

Model	AUC	SE	95% CI	*p*	
PSA	0.538	0.065	0.441–0.664	ref	-
SOC	0.608	0.062	0.486–0.730	ns	ref
Zinc	0.694	0.058	0.581–0.808	ns	ns
Zinc + SOC	0.764	0.052	0.662–0.685	0.002	0.009

## Data Availability

The data presented in this study are available on request from the corresponding author. The data are not publicly available due to privacy restrictions as it contains personal health information of research participants.

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
