# Peer review of "Urinary Zinc Loss Identifies Prostate Cancer Patients"

_cancers, 2022, doi:10.3390/cancers14215316_

Round 1

Reviewer 1 Report

Thank you for the opportunity to review the manuscript entitled, " Urinary Zinc Loss Identifies Prostate Cancer Patients." The authors present an analysis of 633 men who received an indication for a prostate biopsy procedure. The authors quantified zinc levels in urine after a standardized prostatic massage. The authors found that mean zinc levels were lower in the urine of patients with cancer vs healthy subjects. The authors concluded that urinary zinc levels may represent an early and non-invasive diagnostic biomarker for prostate cancer.  The clinical topic is essential, as early diagnosis of prostate cancer, the most commonly diagnosed cancer in men, is to some extent controversial. If zinc could work as a biomarker that would be great. However, I have several comments to improve the quality of the manuscript.

1.    Could the authors include where the data is from and which years it is from in the abstract? Moreover, could the authors provide information that they are conducting regression analysis and the results from this regression analysis?

2.    Did the authors try to measure zinc in patients where there was no prostate massage?

3.    Were median zinc levels comparable to mean levels? Mean levels assume a standard deviation and get influenced by high outlier values.

4.    In the simple summary, do the authors mean that prostate cancer tissue is known to lose “more” zinc compared to normal prostatic tissue?

5.    It would benefit the paper if the authors included quantitative results on prostate cancer incidence and mortality in the beginning of the abstract.

6.    Could the authors briefly discuss pros and cons of other relevant / similar studies and what makes this study unique compared to those in the introduction?

7.    I don’t fully understand how zinc could be a test to identify men at risk for prostate cancer. Does it require prostate massage? It does not seem feasible to offer this to all men.

8.    Wouldn’t this population be skewed and not representative of the population as all patients have been deemed at high enough risk to undergo prostate biopsy? This could be relevant to add to the limitations

9.    How did the authors define a diagnosis of prostate cancer before stratification?

10. How did authors divide patients into the cohorts? Based on what?

Author Response

  1. Could the authors include where the data is from and which years it is from in the abstract? Moreover, could the authors provide information that they are conducting regression analysis and the results from this regression analysis?

- This information has been added to the abstract. The results of the regression analysis have been added in the new Table 3.

  1. Did the authors try to measure zinc in patients where there was no prostate massage?

- In a preliminary experiment, the amount of zinc in the urine was evaluated after performing the prostate massage or not. In subjects where massage was not performed, the average zinc level was lower than in those who underwent prostate massage. Without performing prostate massage no significant differences were found between healthy individuals and prostate cancer patients. Thus, we have added a new figure (Figure S1) in the supplementary materials.

  1. Were median zinc levels comparable to mean levels? Mean levels assume a standard deviation and get influenced by high outlier values.

- In Table 2 the value of mean, median, and interquartile range of zinc in different patient groups are reported. As the Reviewer pointed out correctly, the mean can be affected by extreme values. We already performed an analysis excluding the outliers. The average zinc value does not undergo substantial variations and the statistical significance between the different groups remains unchanged (data not shown).

  1. In the simple summary, do the authors mean that prostate cancer tissue is known to lose “more” zinc compared to normal prostatic tissue?

- We mean that prostate cancer tissue loses the capability to absorb and secrete Zinc compared to normal prostate tissue. We correct the sentence.

  1. It would benefit the paper if the authors included quantitative results on prostate cancer incidence and mortality in the beginning of the abstract.

- We thank the Reviewer for this appropriate comment. We included this information in the abstract.

  1. Could the authors briefly discuss pros and cons of other relevant/similar studies and what makes this study unique compared to those in the introduction?

- A brief description of the added value of this work compared to the state of the art has been added in the introduction.

  1. I don’t fully understand how zinc could be a test to identify men at risk for prostate cancer. Does it require prostate massage? It does not seem feasible to offer this to all men.

- The need for prostate massage before urine collection could represent a limitation for applying the Zinc analysis to the general male population. However, rectal exploration is a procedure that is an integral part of the urological examination within the diagnostic path of prostate cancer. This aspect means that the analysis can also be carried out at an early stage of the check-up process without introducing further discomfort for the patient.

  1. Wouldn’t this population be skewed and not representative of the population as all patients have been deemed at high enough risk to undergo prostate biopsy? This could be relevant to add to the limitations

- Surely the results obtained in this study cannot be generalized, considering them applicable to the general population. Recruited subjects received a prostate biopsy indication according to the Guidelines. The fact of being subjected to this diagnostic test was mandatory given the need to have a confirmation of the presence or absence of the disease. As reported in Table 1, approximately 50% of subjects undergoing biopsy for a clinical suspicion did not have prostate cancer indicating the need to improve the diagnostic performance of the current standard of care. This study aims to represent a first step to support the idea of ​​evaluating urinary Zinc at an earlier point in the diagnosis process, on a less preselected population. As suggested by the Reviewer, a sentence regarding this limitation has been added to the discussion.

  1. How did the authors define a diagnosis of prostate cancer before stratification?

- The diagnosis of prostate cancer was established by histopathological examination of prostate needle biopsy samples at Pathology department of AOU Città della Salute e della Scienza di Torino hospital. According to the Guidelines, a Gleason Score equal or greater than 6 defines the diagnosis of prostate cancer.

  1. How did authors divide patients into the cohorts? Based on what?

- The recruitment of the participants lasted 4 years and the samples were collected sequentially. The total number of subjects recruited was divided according to the order of recruitment by placing the first two thirds of the subjects in the training cohort and the remaining third in the validation cohort.

Reviewer 2 Report

Dear Authors, I have read with interest your article entitled “Urinary Zinc Loss Identifies Prostate Cancer Patients”. I think it adds some useful information in the panorama of the available tools for prostate cancer diagnosis.

Here reported are some minor limitations to correct:

- There are several punctuation errors throughout the text, please correct

- Please remove/move to other paragraph lines 72-78 (it's not correct to report it in the introduction paragraph)

- How much does this test cost? Is this a test which could be performed in every hospital laboratory?

- Please change the order of the two sentences in conclusion paragraph

Author Response

Thank you for your comments.

  1. There are several punctuation errors throughout the text, please correct

- We have checked and corrected punctuation errors.

  1. Please remove/move to other paragraph lines 72-78 (it's not correct to report it in the introduction paragraph)

- As suggested by the Reviewer, the lines 72-78 were removed from the introduction.

  1. How much does this test cost? Is this a test which could be performed in every hospital laboratory?

- Urine Zinc measurement is already in use in some clinical biochemistry laboratories to detect industrial and accidental exposure to Zinc or malabsorption. The cost reimbursed by the National Health System in Italy for this exam is around 5 euro.

  1. Please change the order of the two sentences in conclusion paragraph

- We have changed the order of the sentences.

Round 2

Reviewer 1 Report

The authors have done a nice job responding to my comments. The revised paper is much easier for me to follow. I do not have additional comments.

Author Response

Thank you for allowing us to improve the level of our article.